# Epstein–Barr Virus and Peri-Implantitis: A Systematic Review and Meta-Analysis

**DOI:** 10.3390/v13020250

**Published:** 2021-02-05

**Authors:** Elisabet Roca-Millan, Judith Domínguez-Mínger, Mayra Schemel-Suárez, Albert Estrugo-Devesa, Antonio Marí-Roig, José López-López

**Affiliations:** 1Faculty of Medicine and Health Sciences (School of Dentistry), University of Barcelona, 08907 Barcelona, Spain; erocamil@gmail.com (E.R.-M.); judominger@gmail.com (J.D.-M.); mayraschemel@gmail.com (M.S.-S.); 2Oral Health and Masticatory System Group-IDIBELL, Faculty of Medicine and Health Sciences (School of Dentistry), Odontological Hospital University of Barcelona, University of Barcelona, 08907 Barcelona, Spain; albertestrugodevesa@gmail.com; 3Department of Maxillofacial Surgery, University Hospital of Bellvitge, 08907 Barcelona, Spain; amari@bellvitgehospital.cat

**Keywords:** peri-implantitis, Epstein–Barr virus, human herpesvirus 4, peri-implant disease, dental implants, peri-implant pathology, viruses

## Abstract

The exponential growth in the use of dental implants in the last decades has been accompanied by an increase in the prevalence of peri-implant disease. It appears that viruses may have pathogenic potential for the development of this pathology. The objective of this systematic review is to study the possible association between the presence of Epstein–Barr virus and the development of peri-implantitis. An electronic search was conducted in PubMed/MEDLINE, Scielo and Embase databases for cross-sectional and case–control studies in humans published up to and including 4 January 2021. Five studies were included in the qualitative analysis. The meta-analysis did not show a statistically significant difference regarding the prevalence of Epstein–Barr virus in the peri-implant sulcus between implants with peri-implantitis and healthy implants. In conclusion, no association between the human herpesvirus 4 and peri-implantitis was found. Further research on this topic is essential to develop more effective treatments.

## 1. Introduction

In the last decades, the use of dental implants has increased significantly, suffering an exponential growth since the first decade of the 21st century [1,2,3]. In fact, a study carried out in the United States found that between 1999 and 2016 the prevalence of dental implants increased by 14% per year and projection estimates suggest that it could be as high as 23% by 2026 [3].

According to the scientific literature, dental implants show survival rates of up to 98% at 10 years [4,5,6]. However, the higher use of dental implants has been accompanied by an increase in the prevalence of peri-implant pathology [7]. A recent systematic review and meta-analysis reported an incidence of peri-implantitis ranging from 0.4% over 3 years, to 43.9% within 5 years after implant restoration [8].

Besides minor prosthetic complications (such as crown loosening or ceramic chipping), which are easy to solve, peri-implantitis is the most frequent complication and its resolution is a real challenge, the reason why so much has been published in recent years on this topic [9]. This pathology is defined clinically as the presence of signs of bleeding and/or suppuration in the peri-implant tissues on probing, increased probing depth and loss of 2 or more millimetres of marginal bone compared to previous radiographic recordings [7,10]. Its appearance is the result of a discrepancy between the bacterial challenge and the host response [11]. A large number of risk factors and indicators have been associated with peri-implant disease, like smoking or diabetes mellitus [12,13,14,15], but bacterial infection plays the most important role in its development [8,16,17]. Among the strains most frequently related to peri-implant disease there can be found *Aggregatibacter actinomycetemcomitans*, *Prevotella intermedia*, *Tannerella forsythia*, *Treponema denticola*, *Campylobacter rectus*, *Treponema socranskii*, *Porphyromonas gingivalis*, *Staphylococcus aureus,* and *Campylobacter gracilis* [7,9,18]. However, it has been observed that other pathogens such as *Candida* spp. or some viruses have an important pathogenic potential in the appearance and evolution of peri-implant disease [7,19,20]. Growing evidence suggests that Epstein–Barr virus (EBV) plays a role in the pathogenesis of periodontitis and peri-implantitis [21]. EBV is an enveloped herpesvirus with double-stranded DNA, and it is estimated to infect more than 90% of the adult population [19]. It is known that EBV is transmitted from host to host by salivary contact and the virus passes through the oropharyngeal epithelium to B lymphocytes, where it establishes a lifelong latent infection [19,22,23,24].

EBV can induce inflammatory process, being related to autoimmune diseases like systemic lupus erythematosus, rheumatoid arthritis and multiple sclerosis [25] and is nowadays associated with 1% of global cancers, mostly lymphomas and carcinomas [24].

The hypothesis than EBV may be related to peri-implant disease appears when the presence of EBV is discovered in a high percentage in subgingival plaque samples from periodontal patients [26] or in the peri-implant sulcus [18,21]. Additionally, reduction in subgingival EBV levels following periodontal treatment was demonstrated as well [27]. Therefore, the virus may deteriorate immunologic stability in periodontal/peri-implant disease by contributing to the overgrowth and aggressiveness of inflammophilic bacterial periopathogens, thus favoring the initiation and progression of peri-implant tissues breakdown [27,28]. Actually, the EBV and bacterial periopathogens interaction is bidirectional [29], as specific anaerobic bacteria structural components have the ability to stimulate EBV and the virus may influence pathogenic bacteria overgrowth by affecting potential adhesion to infected host cells and altering the inflammatory cells involved in the immune response [30,31].

The objective of the present systematic review is to study the possible association between EBV and the development of peri-implantitis.

## 2. Materials and Methods

This systematic review was conducted according to the guidelines of the Preferred Reporting Items of Systematic Reviews and Meta-Analyses (PRISMA) statement [32].

### 2.1. PECO Question (Population, Exposure, Comparison, Outcome)

P: Patients with at least one dental implant.

E: Peri-implantitis.

C: Healthy implants.

O: Prevalence of Epstein–Barr virus.

### 2.2. Eligibility Criteria

Studies were eligible for inclusion if they met the following criteria:Cross-sectional studies and case–control studies.Written in English or Spanish.That evaluated the correlation between Epstein–Barr virus and the presence of peri-implantitis.Minimum sample of 20 implants.Peri-implantitis implants group (test) and healthy implants group (control).Peri-implant health diagnosis: radiological evaluation of bone loss and assessment of at least one clinical parameter.Systemically healthy subjects.No history of antibiotic therapy for at least three months.

### 2.3. Search Strategy

An electronic search was conducted by two reviewers (E.R.-M. and J.D.-M.) in MEDLINE (PubMed), Embase and Scielo on 4 January 2021 for articles published up to that date. An additional hand search was performed to identify potential articles of interest in the references of the studies found in the electronic search. The following term combination was used: (“epstein barr virus [All Fields]” OR “human herpesvirus 4 [All Fields]”) AND (“peri-implantitis [All Fields]” OR “peri-implant diseases [All Fields]”).

### 2.4. Study Selection

After discarding duplicate articles, titles and abstracts were read to verify that the identified articles met the inclusion criteria. Finally, the full text of the selected articles was read to corroborate that they met the inclusion and exclusion criteria. Disagreements during the study selection were solved by consulting a third author (J.L.-L.). 

### 2.5. Data Extraction

Data were collected by an author (E.R.-M.) and entered into a data collection form and later they were verified by a second author (J.L.-L.). The following data were extracted: author(s), year of publication, type of study, number of total dental implants, number of healthy implants, number of peri-implantitis implants, number of mucositis implants, Epstein–Barr virus prevalence, sample origin, and main findings.

### 2.6. Quality Assessment

The Newcastle–Ottawa Quality Assessment Form for Case–Control Studies and the Newcastle–Ottawa Quality Assessment Form Adapted for Cross-sectional Studies were implemented to evaluate the methodological quality of the included articles [33]. Two authors (E.R.-M. and J.D.-M.) independently assessed the studies and any discrepancies were solved consulting a third author (J.L.-L.). The Newcastle–Ottawa Scale (NOS) judges the quality considering three domains: selection of study groups, comparability of the groups and outcome/exposure. The maximum score is 9 points for case–control studies and 10 points for cross-sectional studies.

### 2.7. Statistical Analysis

Review Manager (RevMan) (computer program, version 5.4, The Cochrane Collaboration, 2020) was used to evaluate the association between the presence of EBV in the peri-implant sulcus and the peri-implant tissues health. The Mantel–Haenszel random-effects model was implemented in the meta-analysis. The level of significance was set at *p* < 0.05. Heterogeneity was assessed with Chi^2^ and I^2^ tests.

## 3. Results

### 3.1. Study Selection

A total of 28 studies were identified through the electronic and manual searches. Half of them were excluded due to duplication. Of these 14 articles, five were discarded after reading the titles and three after reading the abstract as they did not meet the inclusion criteria. Of the six full-text articles with potential for inclusion [18,19,21,27,34,35], one was excluded as it appeared to be the same study as another included paper but in earlier stage [18]. Finally, a total of five studies were included in the qualitative analysis [19,21,27,34,35] (Figure 1).

### 3.2. Study Methods and Characteristics

Three of the included articles were cross-sectional studies [21,27,35] and the other two were case–control studies [19,34] (Table 1). The papers were published between 2011 and 2018. Three of them were conducted in training centers [19,21,27], one in a private dental clinic [35] and another one does not specify where it took place [34]. A total of 274 patients (149 women and 125 men) and 388 implants (197 healthy implants, 166 peri-implantitis implants and 25 mucositis implants) were included in the analysis. In all the studies samples were taken to perform a real-time polymerase chain reaction (PCR) to detect the presence of EBV. In two of the articles only subgingival plaque samples of the peri-implant tissues were extracted [19,21], in another one the source of the sample was saliva [34], in another samples were taken both from saliva and subgingival peri-implant tissues [35] and in the last one samples were extracted from subgingival plaque and from the internal implant connection [27]. Only one study differentiated between EBV genotypes (EBV-1 and EBV-2) [21].

### 3.3. Quality Assessment

All the studies were assessed using the Newcastle–Ottawa Scale (NOS) [33]. The comparability was the dimension that received the lowest score, while the evaluation of the outcome/exposure was the one that obtained the highest (Table 2). The mean NOS score was 6.6 (±1.34). Two of the studies can be considered of good quality [21,27] and the other three of satisfactory quality [19,34,35].

### 3.4. EBV and Peri-Implantitis

In three of the included studies, no statistically significant difference was observed in the prevalence of EBV between patients with peri-implantitis (PI) and patients with healthy implants (HI) [19,27,34]. However, in two of them the PI group presented higher prevalence of EBV [19,34], with no difference in the third study [27].

In the other two works [21,35], statistically significant differences were found in the prevalence of the virus between both groups. The group with PI showed a significantly higher prevalence of EBV compared to the HI group. Jankovic et al. [21] observed this result for EBV-1 genotype not for EBV-2 genotype.

Only one of the studies included also subjects with mucositis, a reversible peri-implant pathology characterized by gingival inflammation without associated bone loss. In that article a statistically significant correlation between the presence of EBV-1 and mucositis was also found [21].

In cases where samples were extracted from two different sites, more EBV positives were obtained from subgingival plaque above the saliva [35] or the internal implant connection [27].

In relation to the association between EBV and other periopathogens, significantly higher median loads of *P. intermedia* and *C. rectus* were found in EBV positives compared to EBV negatives [27]. Furthermore, the coexistence between *P. gingivalis* and EBV was statistically significantly higher in patients with peri-implantitis [19].

Regarding the inflammatory response, a statistically significant association was found between the presence of herpesvirus and the presence in saliva of the markers macrophage inflammatory protein-1β (MIP-1β) and tumor necrosis factor-α (TNF-α) in the PI group [34].

### 3.5. Quantitative Analysis

Studies in which subgingival plaque samples were taken were selected for the quantitative analysis [19,21,27,35]. No statistically significant difference was found regarding the presence of EBV in the peri-implant sulcus between PI and HI groups (OR = 4.14; 95% CI: 0.93–18.37; z = 1.87; *p* = 0.06) (Figure 2). Heterogeneity tests from pooled showed statistical significance (Chi-squared = 11.96, *p* < 0.008) (Figure 2). 

## 4. Discussion

In the present meta-analysis no statistically significant difference was found regarding the presence of EBV in the peri-implant sulcus between implants with peri-implantitis and healthy implants. However, it must be considered that due to the limited literature published on this topic, only four articles [19,21,27,35] could be included in the quantitative analysis and the one with a larger sample and therefore greater weight is the study in which no differences between groups were observed [27].

Only one of the studies evaluated both EBV genotypes, obtaining significantly higher prevalence of the EBV-1 genotype in the PI group, which was not the case with the EBV-2 genotype [21]. This suggests that this genotype could be more present in the development of peri-implant pathology. According to the same study, the development of mucositis could already be encouraged by the presence of EBV-1 [21].

The higher prevalence of the virus in the peri-implant sulcus than in saliva of the same subjects highlights the possible role of EBV in peri-implant diseases [35]. Furthermore, the coexistence of the virus with periopathogenic bacteria such as *P. gingivalis* [19], *P. intermedia* or *C. rectus* [27] could be a key point in the development of peri-implantitis.

Regarding the pathogenesis, it seems that the presence of EBV could be related to an increase in the inflammatory response with the consequent increase in markers such as the macrophage inflammatory protein-1β (MIP-1β) and tumor necrosis factor-α (TNF-α) [34].

There is no review so far that specifically focused on the relationship between EBV and peri-implantitis, but there are three recent reviews that study the microbiological profile of peri-implantitis [36] or the association of viruses [37], or more explicitly the herpesvirus [38], with the appearance of this pathology.

One of these reviews concluded that the microbiological profile in peri-implantits consists of Gram-negative anaerobic periopathogens and opportunistic microorganisms almost in the same proportion and that is often associated with EBV [36]. Another review concluded that the presence of herpesvirus in the peri-implant subgingival biofilm is an indicator of peri-implant disease [38].

Unlike the present review, the only existing meta-analysis did find a statistically significant difference in the presence of EBV between the PI and HI groups. However, this difference between the two meta-analyses is based solely on the inclusion of a single different article in the quantitative analysis [37].

The main limitation of this review is the limited scientific literature published on this topic, which makes it impossible to draw conclusions about the possible association between EBV and peri-implantitis. Furthermore, some of the included studies analyze small samples or may represent a source of bias, especially due to the lack of information on comparability between groups.

Future research in this area should focus on case–control studies, with large samples, with a higher methodological quality to reduce the risk of bias and in which samples were obtained from the peri-implant sulcus and not from saliva or from the implant connection.

## 5. Conclusions

No statistically significant difference was found in terms of the prevalence of EBV in the peri-implant sulcus between PI and HI groups. More case–control studies of high methodological quality are necessary to evaluate the possible association between EBV and peri-implantitis. Due to the high prevalence of periodontal disease and its difficult treatment, it is essential to continue researching this topic, since better knowledge of its pathogenesis will allow the development of more effective therapies. 

## Figures and Tables

**Figure 1 viruses-13-00250-f001:**
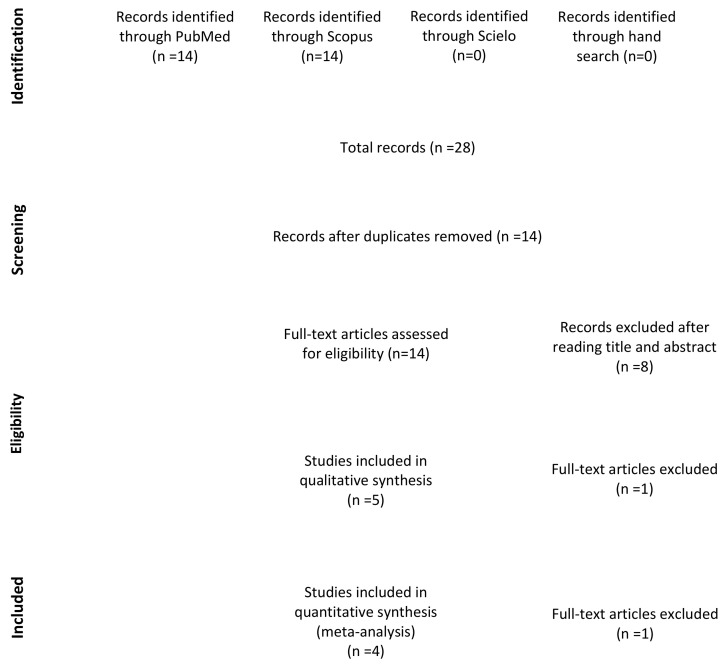
Preferred Reporting Items for Systematic Reviews and Meta-Analyses (PRISMA) flow diagram of selection process.

**Figure 2 viruses-13-00250-f002:**
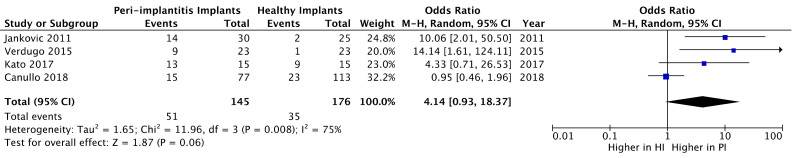
Forest plot of the prevalence of Epstein–Barr virus (EBV) in peri-implantitis implants versus healthy implants.

**Table 1 viruses-13-00250-t001:** Summary of the included studies.

Author	Type of Study	N Implants				EBV in HI	EBV in PI	EBV in MI	Sample
		Healthy Implants	Peri-Implantitis	Mucositis	Total				
Jankovic et al., 2011 [21]	Cross-sectional	25	30	25	90	0/25 (0%) EBV-1; 2/25 (8%) EBV-2	11/30 (36.7%) EBV-1; 3/30 (10%) EBV-2	8/25 (32%) EBV-1; 1/25 (4%) EBV-2	Subgingival plaque
Verdugo et al., 2015 [35]	Cross-sectional	23	23	0	46	1/23 (4.3%)	9/23 (39.1%)	x	Subgingival plaque and saliva
Kato et al., 2017 [19]	Case-control	15	15	0	30	9/15 (60%)	13/15 (86.7%)	x	Subgingival plaque
Canullo et al., 2018 [27]	Cross-sectional	113	77	0	190	23/113 (20.35%)	15/77 (19.5%)	x	Subgingival plaque and internal implant connection
Marques et al., 2018 [34]	Case-control	21	21	0	42	2/21 (9.5%)	4/21 (19%)	x	Saliva
Total		197	166	25	388				

Abbreviations: EBV, Epstein–Barr virus; HI, healthy implants; MI, mucositis implants; PI, peri-implantitis implants.

**Table 2 viruses-13-00250-t002:** Quality assessment of the included studies according to the Newcastle–Ottawa Scale (NOS).

	Selection	Comparability	Outcome/Exposure
Jankovic et al., 2011 [21]	⋆⋆⋆	⋆⋆	⋆⋆⋆
Verdugo et al., 2015 [35]	⋆⋆⋆		⋆⋆⋆
Kato et al., 2017 [19]	⋆⋆	⋆	⋆⋆⋆
Canullo et al., 2018 [27]	⋆⋆⋆	⋆⋆	⋆⋆⋆
Marques et al., 2018 [34]	⋆⋆	⋆	⋆⋆

* A star is awarded for each item within the Selection and Outcome/Exposure categories. A maximum of two stars can be given for Comparability.

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
