# Peer review of "Epstein–Barr Virus and Peri-Implantitis: A Systematic Review and Meta-Analysis"

_viruses, 2021, doi:10.3390/v13020250_

Round 1
Reviewer 1 Report
The paper systematically reviews the possible relationship between the presence of Epstein-Barr virus and peri-implantitis. The studied topic is interesting and the review has good methodological quality. In my opinion, this manuscript can be accepted after the following minor revisions:
- For the sentence "According to the scientific literature, dental implants have survival rates of up to 99% at 10 years", in lines 35 and 36, more appropriate references should be given.
- In section 2.5, lines 155-158, the number of patients with EBV does not appear as extracted data. It should be added since this information is included in the table.
- In section 3.3, you should add a global judgment on the risk of bias of the included articles.
- I suggest that the discussion does not include subsections.
Reviewer 2 Report
This manuscript requires some minor modifications.
Page 1 lines 37,38: there are typos.
Figure 1 can be deleted, it is not useful.
Reference 4 is not appropriate. A recent implant survival study could be this:
Cassetta M, Di Mambro A, Giansanti M, Brandetti G. The Survival of Morse Cone-Connection Implants with Platform Switch. Int J Oral Maxillofac Implants. 2016 Sep-Oct;31(5):1031-9. doi: 10.11607/jomi.4225. PMID: 27632257.
